# Involvement of TRPV4 in temperature-dependent perspiration in mice

**Makiko Kashio[1,2,3†], Sandra Derouiche[1†], Reiko U Yoshimoto[4], Kenji Sano[5‡], Jing Lei[1,2,6], Mizuho A Kido[4], Makoto Tominaga[1,2,7]\***

[1]Division of Cell Signaling, National Institute for Physiological Sciences, National Institutes of Natural Sciences, Okazaki, Japan; [2]Thermal Biology Group, Exploratory Research Center on Life and Living Systems, National Institutes of Natural Sciences, Okazaki, Japan; [3]Department of Cell Physiology, Faculty of Life Sciences, Kumamoto University, Kumamoto, Japan; [4]Division of Histology and Neuroanatomy, Department of Anatomy and Physiology, Faculty of Medicine, Saga University, Saga, Japan; [5]Department of Laboratory Medicine, Shinshu University Hospital, Matsumoto, Japan; [6]Department of Dermatology, Graduate School of Medicine, Osaka University, Suita, Japan; [7]Thermal Biology Research Group, Nagoya Advanced Research and Development Center, Nagoya City University, Nagoya, Japan

**\*For correspondence:**
tominaga@nips.ac.jp

[†]These authors contributed equally to this work

**Present address:** [‡]Department of Pathology, Iida Municipal Hospital, Iida, Japan

**Competing interest:** The authors declare that no competing interests exist.

**Abstract** Reports indicate that an interaction between TRPV4 and anoctamin 1 (ANO1) could be widely involved in water efflux of exocrine glands, suggesting that the interaction could play a role in perspiration. In secretory cells of sweat glands present in mouse foot pads, TRPV4 clearly colocalized with cytokeratin 8, ANO1, and aquaporin-5 (AQP5). Mouse sweat glands showed TRPV4-dependent cytosolic $Ca^{2+}$ increases that were inhibited by menthol. Acetylcholine-stimulated sweating in foot pads was temperature-dependent in wild-type, but not in TRPV4-deficient mice and was inhibited by menthol both in wild-type and TRPM8KO mice. The basal sweating without acetyl-choline stimulation was inhibited by an ANO1 inhibitor. Sweating could be important for maintaining friction forces in mouse foot pads, and this possibility is supported by the finding that wild-type mice climbed up a slippery slope more easily than TRPV4-deficient mice. Furthermore, TRPV4 expression was significantly higher in controls and normohidrotic skin from patients with acquired idiopathic generalized anhidrosis (AIGA) compared to anhidrotic skin from patients with AIGA. Collectively, TRPV4 is likely involved in temperature-dependent perspiration via interactions with ANO1, and TRPV4 itself or the TRPV4/ANO 1 complex would be targeted to develop agents that regulate perspiration.

## eLife assessment

This **useful** studying implicates TRPV4 as a mediator of sweat, potentially based on TRPV4's expression and function on sweat glands. The data and methods are **solid**, with some limitations in terms of the approach. Overall, the work lends new insight into the physiological basis of sweating using data from mice and humans.

## Introduction

Sweating is a vital physiological process (*Shibasaki and Crandall, 2010*). There are two basic types of sweating: thermoregulatory and emotional sweating, in addition to gustatory sweating, largely localized to the face and neck regions, that occurs while consuming some foods, particularly pungent foods (*Lee, 1954*). Most sweat glands are of the eccrine type, and they produce a thin secretion that

**eLife digest** Stress, spicy foods and elevated temperatures can all trigger specialized gland cells to move water to the skin – in other words, they can make us sweat. This process is one of the most important ways by which our bodies regulate their temperature and avoid life-threatening conditions such as heatstroke. Disorders in which this function is impaired, such as AIGA (acquired idiopathic generalized anhidrosis), pose significant health risks. Finding treatments for sweat-related diseases requires a detailed understanding of the molecular mechanisms behind sweating, which has yet to be achieved.

Recent research has highlighted the role of two ion channels, TRPV4 and ANO1, in regulating fluid secretion in glands that produce tears and saliva. These gate-like proteins control how certain ions move in or out of cells, which also influences water movement. Once activated by external stimuli, TRPV4 allows calcium ions to enter the cell, causing ANO1 to open and chloride ions to leave. This results in water also exiting the cell through dedicated channels, before being collected in ducts connected to the outside of the body.

TRPV4, which is activated by heat, is also present in human sweat gland cells. This prompted Kashio et al. to examine the role of these channels in sweat production, focusing on mice as well as AIGA patients. Probing TRPV4, ANO1 and AQP5 (a type of water channel) levels using fluorescent antibodies confirmed that these channels are all found in the same sweat gland cells in the foot pads of mice. Further experiments highlighted that TRPV4 mediates sweat production in these animals via ANO1 activation.

As rodents do not regulate their body temperature by sweating, Kashio et al. explored the biological benefits of having sweaty paws. Mice lacking TRPV4 had reduced sweating and were less able to climb a slippery slope, suggesting that a layer of sweat helps improve traction.

Finally, Kashio et al. compared samples obtained from healthy volunteers with those from AIGA patients and found that TRPV4 levels are lower in individuals affected by the disease. Overall, these findings reveal new insights into the underlying mechanisms of sweating, with TRPV4 a potential therapeutic target for conditions like AIGA. The results also suggest that sweating could be controlled by local changes in temperature detected by heat-sensing channels such as TRPV4. This would depart from our current understanding that sweating is solely controlled by the autonomic nervous system, which regulates involuntary bodily functions such as saliva and tear production.

is hypotonic to plasma. Although eccrine sweat glands are distributed all over the body, their density is highest in the axillary region and on the palms of the hands and the soles of the feet. In humans, the main function of eccrine sweat glands is body temperature regulation. Meanwhile, apocrine sweat glands are found primarily in the axillae and urogenital regions. These scent glands become active during puberty and secrete a viscous fluid that is associated with body odor.

Body temperature regulation is important to maintain homeostasis. Body temperature is poorly controlled in patients with hypohidrosis. Meanwhile, patients affected by hyperhidrosis can have difficulty in social and professional situations due to increased sweat production, and the resulting subjective perception of illness at an individual level may be substantial (*Cohen and Solish, 2003*; *Schlereth et al., 2009*). However, the molecular mechanisms of perspiration are not clearly understood.

We previously reported the functional interaction between TRP channels and the $Ca^{2+}$-activated $Cl^-$ channel, anoctamin 1 (ANO1, also known as TMEM16A) (*Takayama et al., 2014*; *Takayama et al., 2015*; *Derouiche et al., 2018*). TRP channels have high $Ca^{2+}$ permeability (*Gees et al., 2012*), and $Ca^{2+}$ entering cells through TRP channels activates ANO1 by making a physical complex, leading to $Cl^-$ efflux in cells with high intracellular $Cl^-$ concentrations. The $Cl^-$ efflux may drive water efflux through water channels in exocrine gland acinar cells that increase exocrine function and causes depolarization in primary sensory neuron that increases nociception. For skin keratinocytes that have relatively low intracellular $Cl^-$ concentrations, interaction between TRPV3 and ANO1 causes $Cl^-$ influx, followed by increased cellular movement/proliferation in response to cell cycle modulation (*Yamanoi et al., 2023*). Thus, direction of $Cl^-$ movement through ANO1 is simply determined by the balance between equilibrium potentials of $Cl^-$ and membrane potentials in each cell (*Takayama et al., 2019*).

The involvement of TRPV4 in exocrine gland function prompted us to examine the functional interaction in perspiration because TRPV4 is expressed in human eccrine sweat glands (*Delany et al., 2001*). Although sweat glands are innervated by sympathetic neurons, acetylcholine (Ach) is released from the nerve endings (*Hu et al., 2018*). We show that the functional interaction of TRPV4 and ANO1 is involved in temperature-dependent sweating and increased friction force.

## Results

### Expression of TRPV4, anoctamin 1, and AQP5 in mouse sweat glands

We detected expression of TRPV4, ANO1, and the water channel aquaporin-5 (AQP5) in the eccrine glands of mouse foot pads. The secretory coil is located in the deep dermis and a relatively straight duct opens to the skin surface. We first validated an anti-TRPV4 antibody that we generated. This anti-TRPV4 antibody conspicuously labeled the basal layer of the epidermis, secretory eccrine gland cells, and duct cells only in skin from wild-type (WT) mice, but not in skin from TRPV4-deficient (TRPV4KO) mice (*Figure 1A*), indicating the antibody specificity. TRPV4 was clearly localized in secretory glands as confirmed by positivity for cytokeratin 8 (CK8), a secretory cell marker (*Figure 1B*). The duct cells were not labeled by ANO1 and CK8 (*Figure 1B*). TRPV4-immunoreactivity was stronger in duct cells near the secretory region and gradually diminished in the distal excretory ducts toward the epidermis. Bilayered sweat ducts showed TRPV4 labeling in basal cells but not suprabasal cells (*Figure 1C*). Secretory cells in human eccrine glands are classified into two types: clear cells that mainly secrete water and electrolytes, and dark cells that secrete macromolecules like glycoproteins. We found that TRPV4-expressing secretory cells were positive for the calcitonin gene-related peptide (CGRP), a dark cell marker, and were heterogeneously labeled (*Figure 1D*). This result is consistent with earlier studies showing that mouse eccrine glands have a more primitive structure than human glands and have only one type of secretory cell that resembles human clear cells but also has dark cell characteristics (*Kurosumi and Kurosumi, 1970*; *Bovell, 2018*).

To explore TRPV4 subcellular localization, we observed tissues using Airyscan super-resolution imaging. TRPV4 was heterogeneously labeled in the gland cells and showed apparent localization in basal and apical membranes (*Figure 1D*). TRPV4 was absent in myoepithelial cells. Conspicuous co-labeling of TRPV4 and ANO1 or AQP5 with filamentous actin (F-actin) was seen at the apical site (luminal side) of the secretory cells (*Figure 1E*). These close topological relationships clearly suggest that TRPV4, ANO1, and AQP5 would be able to form a complex that promotes sweat secretion in eccrine glands of mouse foot pads. These results also suggest that TRPV4-expressing secretory cells are involved in the secretion of macromolecular components as well as the secretion of water and ions.

### Functional expression of TRPV4 in acinar cells of mouse sweat glands

We previously showed the functional interaction of TRPV4 and ANO1 in the heterologous expression with HEK293T cells and mouse native exocrine gland acinar cells (*Takayama et al., 2014*; *Derouiche et al., 2018*). Then, we examined functional TRPV4 expression in sweat glands in mouse foot pads. WT mouse sweat glands responded to the TRPV4 agonist, GSK (500 nM), and to Ach (10 µM) (*Figure 2A*). No cytosolic $Ca^{2+}$ increase induced by GSK was observed in sweat glands from TRPV4KO mice (*Figure 2B*). Interestingly, the GSK-induced increase in cytosolic $Ca^{2+}$ was significantly inhibited by menthol (5 µM) in WT mouse sweat glands, suggesting that menthol inhibited TRPV4 function. Meanwhile, menthol alone caused no change in cytosolic $Ca^{2+}$ concentration (*Figure 2C*). These data indicate functional expression of TRPV4 in mouse secretory cells.

### TRPV4 involvement in perspiration in mice

To examine the functional interaction between TRPV4 and ANO1 in mouse sweat glands in vivo, stimulated sweating induced by Ach (100 µM, 2 min) in mouse hind paws at 25 and 35°C was investigated using an iodine and starch reaction to measure secreted amylase (*Nejsum et al., 2002*). At 25°C, no difference in stimulated sweating was seen both in WT and TRPV4KO mice while sweating was increased at 35°C in WT, but not in TRPV4KO mice (*Figure 3A*). Temperature-dependent basal sweating without Ach stimulation for 15 min was also observed in WT mice, but not in TRPV4KO mice (*Figure 3B*). Menthol, a well-known TRPM8 activator, inhibits ANO1 function (*Takayama et al., 2017*). The ability of menthol to inhibit both TRPV4 and ANO1 suggests that menthol would inhibit sweating.

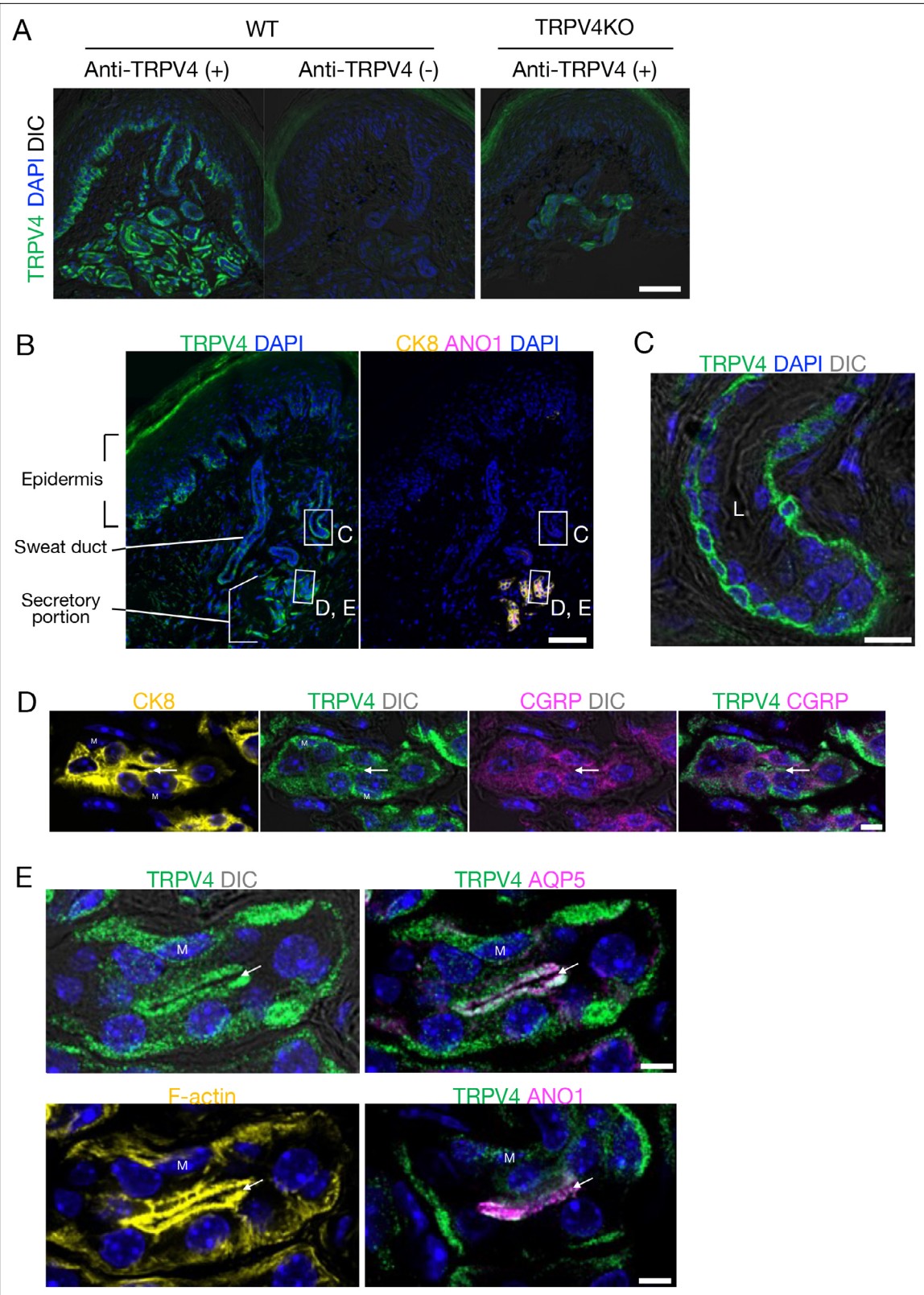

**Figure 1.** TRPV4 localization in eccrine glands of mouse foot pads. (**A**) TRPV4 signals in the secretory coil in the deep dermis with a relatively straight duct opening to the skin surface. (**B**) Localization of TRPV4 (green), cytokeratin 8 (CK8; yellow), and anoctamin 1 (ANO1; magenta) in the skin. (**C–E**) Highly magnified Airyscan super-resolution images of the sweat duct (**C**) and secretory portion (**D, E**). (**C**) TRPV4 localizes to the basal cells of the bilayered sweat duct. Ductal lumen: L. (**D**) Secretory gland showing labeling for TRPV4 and calcitonin gene-related peptide (CGRP). (**E**) Secretory gland

*Figure 1 continued on next page*

*Figure 1 continued*

with conspicuous TRPV4 labeling in myoepithelial cells (M) and secretory cells. TRPV4 clearly colocalizes with aquaporin-5 (AQP5), F-actin, and ANO1 at the luminal side of the secretory cells. Arrows indicate the glandular lumen. DIC, differential interference contrast; nuclei: DAPI. Scale bar: 50 μm (**A, B**). 5 μm (**C–E**).

Accordingly, we compared stimulated sweating with either an ethanol vehicle (used for menthol dilution) or menthol treatment for 2 min. Menthol treatment caused a significantly lower degree of sweating than ethanol treatment both in WT and TRPM8KO mice (*Figure 3C and D*). This result could indicate that menthol inhibits sweating by inhibiting the function of ANO1, independently of TRPM8. Next, in order to prove the involvement of ANO1 in basal sweating, we examined the effects of a strong and specific ANO1 antagonist, Ani9. An Ani9 treatment almost completely abolished basal sweating both at 25 and 35°C (*Figure 3E and F*), indicating a pivotal role of ANO1 in sweating.

## Physiological significance of TRPV4-mediated sweating

Mice do not sweat to control body temperature, so the physiological significance of hind paw sweating is unclear. In humans, fingertip moisture is known to be optimally modulated during object manipulation through regulation of friction force (*André et al., 2010*). The same mechanism might promote the traction of hind paws when mice climb slippery slopes. Here, we constructed a slope covered with slippery vinyl (*Figure 4A*) and compared the climbing behaviors of WT and TRPV4KO mice for 1 hr at 26–27°C with 35–50% humidity. The total number of climbing attempts was the same for WT and TRPV4KO mice (25.6 ± 2.5 for WT, n = 5; 24.7 ± 3.9 for TRPV4KO, n = 4) (*Figure 4B*), but a higher percentage of WT mice successfully climbed to the top of the slope than did TRPV4KO mice (79.5 ± 6.4% for WT; 41.8 ± 2.8% for TRPV4KO; p<0.01) (*Figure 4C and D*, *Videos 1 and 2*). WT mice easily came down the slippery slope. These data suggest that WT mice might produce more hind paw sweat (*Figure 3*) that increase traction on the slope.

## TRPV4 expression in human sweat glands

We next examined whether TRPV4 also plays a role in human perspiration. Patients with acquired idiopathic generalized anhidrosis (AIGA) have acquired impairment in total body sweating even when exposed to heat or engaging in exercise (*Nakazato et al., 2004*; *Munetsugu et al., 2017*; *Sano et al., 2017*). We compared TRPV4 expression in sweat glands from patients with melanocytic nevus (n = 10, ages; 15–63) as controls and patients with AIGA (n = 10, ages; 24–55) using a commercially available anti-TRPV4 antibody different from the one utilized in the mouse samples. All patients with AIGA were male, which is consistent with the gender distribution of AIGA, while

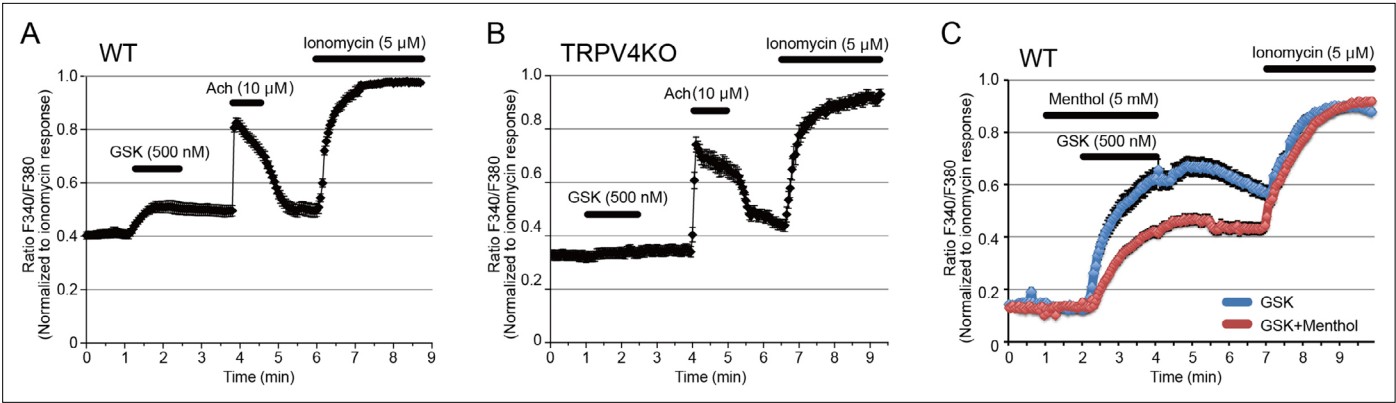

**Figure 2.** Functional TRPV4 expression in mouse sweat gland acinar cells. (**A, B**) Changes in cytosolic $Ca^{2+}$ concentrations upon stimulation with GSK, acetylcholine, or ionomycin in sweat gland acinar cells from wild-type (WT, **A**) and TRPV4-deficient (TRPV4, **B**) mice. n = 6 experiments for WT and TRPV4KO sweat glands. (**C**) Changes in cytosolic $Ca^{2+}$ concentration upon stimulation with GSK in the presence (red) or absence (blue) of menthol in sweat gland acinar cells from WT mice.

The online version of this article includes the following source data for figure 2:

**Source data 1.** Source data files for *Figure 2*.

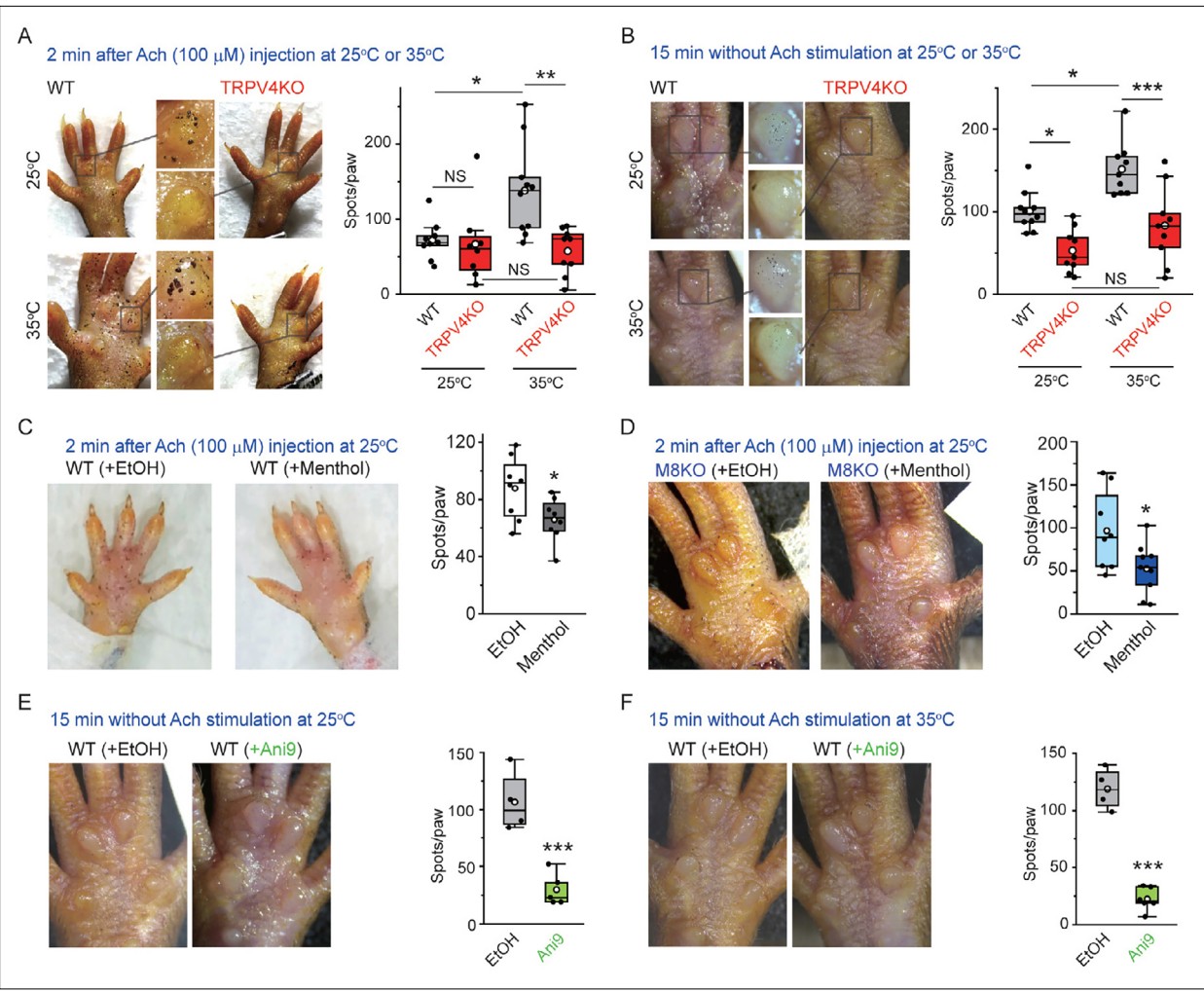

**Figure 3.** Stimulated sweating in mouse hind paws at different temperatures. (**A**, left) Representative stimulated sweat spots formed at 25 or 35°C in hind paws of wild-type (WT) and TRPV4KO mice 2 min after injection of acetylcholine. (**A**, right) Comparison of sweat spots/paws at 25 or 35°C in WT and TRPV4KO mice (box–whisker plot). n = 8–10 for WT or TRPV4KO. $^{NS}$p≥0.05, *p<0.05, **p<0.01 between indicated pairs. (**B**, left) Representative sweat spots at 25 or 35°C in hind paws of WT and TRPV4KO mice without acetylcholine stimulation at 15 min. (**B**, right) Comparison of sweat spots/paw at 25 or 35°C for WT and TRPV4KO mice (box–whisker plot). n = 9–10 for WT or TRPV4KO. $^{NS}$p≥0.05, *p<0.05, ***p<0.001 between indicated pairs. (**C, D**, left) Representative stimulated sweat spots at 25°C in hind paws of WT (**C**) and TRPM8KO (**D**) mice 2 min after injection of acetylcholine with or without menthol. (**C, D**, right) The effect of menthol on sweat spots/paw at 25°C in WT (**C**) and TRPM8KO (**D**) mice (box–whisker plot). n = 8–9 for with or without menthol. *p<0.05. (**D, E**, left) Representative sweat spots without acetylcholine stimulation at 25°C (**E**) and 35°C (**F**) in hind paws of WT mice with or without Ani9. (**D, E**, right) The effect of Ani9 on sweat spots/paw at 25°C (**E**) and 35°C (**F**) in WT mice (box–whisker plot). n = 4–6 for with or without Ani9. ***p<0.001.

The online version of this article includes the following source data for figure 3:

**Source data 1.** Source data files for *Figure 3*.

2 of the 10 controls were female. Representative TRPV4 staining is shown in *Figure 5A and B*. Although signals for TRPV4 staining were high in normohidrotic skin from a patient with AIGA and were equivalent to those of controls, levels in anhidrotic skin from the same patient with AIGA were very low.

We classified TRPV4 staining intensity from 1+ (low) to 3+ (high). Scores were significantly higher in controls and normohidrotic skin from patients with AIGA (2+ or 3+) than anhidrotic skin from AIGA cases (1+ or 2+) (mean 2.5 ± 0.17 vs. 1.0 ± 0.10 for controls and normohidrotic skin from patients with AIGA vs. anhidrotic skin from AIGA cases, respectively, p<0.0001) (*Figure 5*). These data clearly indicate that TRPV4 plays a role in normal perspiration in humans.

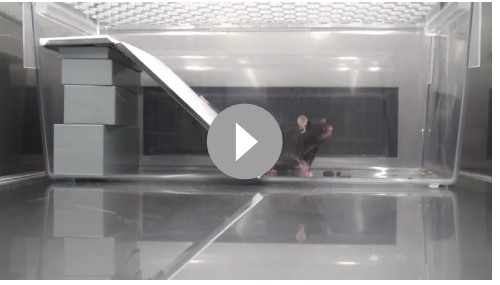

**Figure 4.** Climbing behaviors on a slippery slope. (**A**) A mouse in a cage containing the vinyl slope. (**B**) Number of attempts made by wild-type (WT) and TRPV4KO mice within 60 min. (**C**) Successful (climbing) and failed (slipping) climbing behaviors exhibited by WT and TRPV4KO mice within 60 min. Different colors indicate individual mice. n = 5 for both WT and TRPV4KO. (**D**) Comparison of climbing success rates between WT and TRPV4KO mice. *p<0.05.

**Video 1.** WT mice successfully climbed to the top of the slippery slope and easily came down the slope. https://elifesciences.org/articles/92993/figures#video1

## Discussion

$Ca^{2+}$ entering cells through TRP channels is known to be involved in various $Ca^{2+}$-mediated events, particularly in non-excitable cells, whereas cation influx-induced depolarization is important for excitation of primary sensory neurons through activation of voltage-gated $Na^+$ channels (*Takayama et al., 2014*; *Takayama et al., 2015*; *Derouiche et al., 2018*). $Ca^{2+}$ entering cells is instantaneously chelated to maintain low intracellular $Ca^{2+}$ concentrations. However, high $Ca^{2+}$ conditions can persist for longer periods just beneath

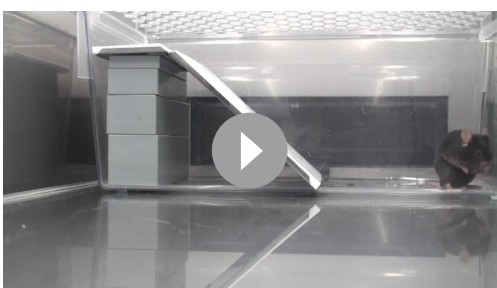

**Video 2.** TRPV4KO mice failed to climb to the top of the slippery slope.
https://elifesciences.org/articles/92993/figures#video2

the plasma membrane. We reported that several TRP channels, including TRPV1, TRPV3, TRPV4, and TRPA1, can form a complex with the $Ca^{2+}$-activated $Cl^-$ channel, ANO1, and activate ANO1 via $Ca^{2+}$ entering cells through TRP channels (*Takayama et al., 2014*; *Takayama et al., 2015*; *Derouiche et al., 2018*) Interaction between TRPV4 and ANO1 causes $Cl^-$ efflux, followed by water efflux, suggesting that the complex could be involved in exocrine gland functions including secretion of cerebrospinal fluid, saliva, and tears (*Takayama et al., 2014*; *Derouiche et al., 2018*). We demonstrated that the TRPV4-ANO1 interaction is also involved in water efflux associated with the exocrine function during sweating in this study. Notably, the TRPV4, ANO1, and AQP5 complex is confined to acinar cells in secretory sweat glands, whereas TRPV4 is also expressed at other sites in skin tissues (*Figure 1*). This result could indicate a specific function for the complex in water efflux occurring in exocrine glands.

It is generally believed that signals in the brain activate sympathetic nerves that cause perspiration by releasing Ach at the sympathetic postganglionic fibers. This is a simple mechanism for body temperature regulation with perspiration. However, sweat glands themselves could sense local heating and cause sweating through warmth-sensitive TRPV4 channel activation that we clarified in this study. This local temperature sensation by TRPV4 could also be true for saliva and tear secretion that involves TRPV4/ANO1 interaction (*Derouiche et al., 2018*).

Several human diseases involve hypohidrosis or hyperhidrosis (*Cohen and Solish, 2003*; *Schlereth et al., 2009*; *Cheshire, 2020*). Patients with hypohidrosis have difficulty regulating body temperature in response to high temperatures and can experience dizziness, muscle cramps, weakness, high fever, or nausea that is typically not serious. However, patients with hypohidrosis sometimes have heatstroke, which is the most serious complication; the incidence of heatstroke has recently increased with global warming. Furthermore, some patients with collagen diseases like Sjögren's syndrome, an autoimmune exocrinopathy, suffer from hypohidrosis as well as dry mouth and dry eye that is not easily treated (*Katayama, 2018*). AIGA is also characterized by hypohidrosis without clear etiology (*Nakazato et al., 2004*; *Munetsugu et al., 2017*; *Sano et al., 2017*). In Japan, both Sjögren's syndrome and AIGA are classified as designated intractable diseases (nos. 53 and 163, respectively). Problems with exocrine gland function in patients with Sjögren's syndrome as well as the low TRPV4 expression levels in patients with AIGA suggest that TRPV4 could be a key molecule involved in these diseases and that novel treatment strategies could target TRPV4 and/or ANO1. There are two recent reports showing that TRPV4 was not critical in regulating sweating in human subjects (*Fujii et al., 2019*; *Fujii et al., 2021*), which is in contradiction to our mouse and human data. The authors focused on the vasodilating effects of TRPV4. We currently have no idea how to explain the apparently different conclusion regarding the involvement of TRPV4. Multiple factors could explain the apparent difference between the two studies. The parameters that they examined are different from ours, and the authors examined human healthy volunteers while we used the sample of patients with AIGA. More investigation would be needed in the future.

The application of menthol to the skin produces a cool, comfortable sensation that is generally thought to result from the activation of the menthol receptor TRPM8. However, the finding that menthol inhibits both TRPV4 and ANO1 suggests that transient reduction in sweating by inhibiting TRPV4 and ANO1 also contributes to the cool sensation. On the other hand, patients with hyperhidrosis can sweat enough to soak their clothing or have sweat drip off their hands (*Cohen and Solish, 2003*; *Schlereth et al., 2009*). Hyperhidrosis can occur as a primary or secondary effect after infections or with some endocrine diseases. Others can experience hyperhidrosis on the palms of their hands when nervous. Development of chemicals targeting TRPV4, ANO1, or the complex could be a new therapeutic strategy for these conditions, for which there are currently no effective treatments.

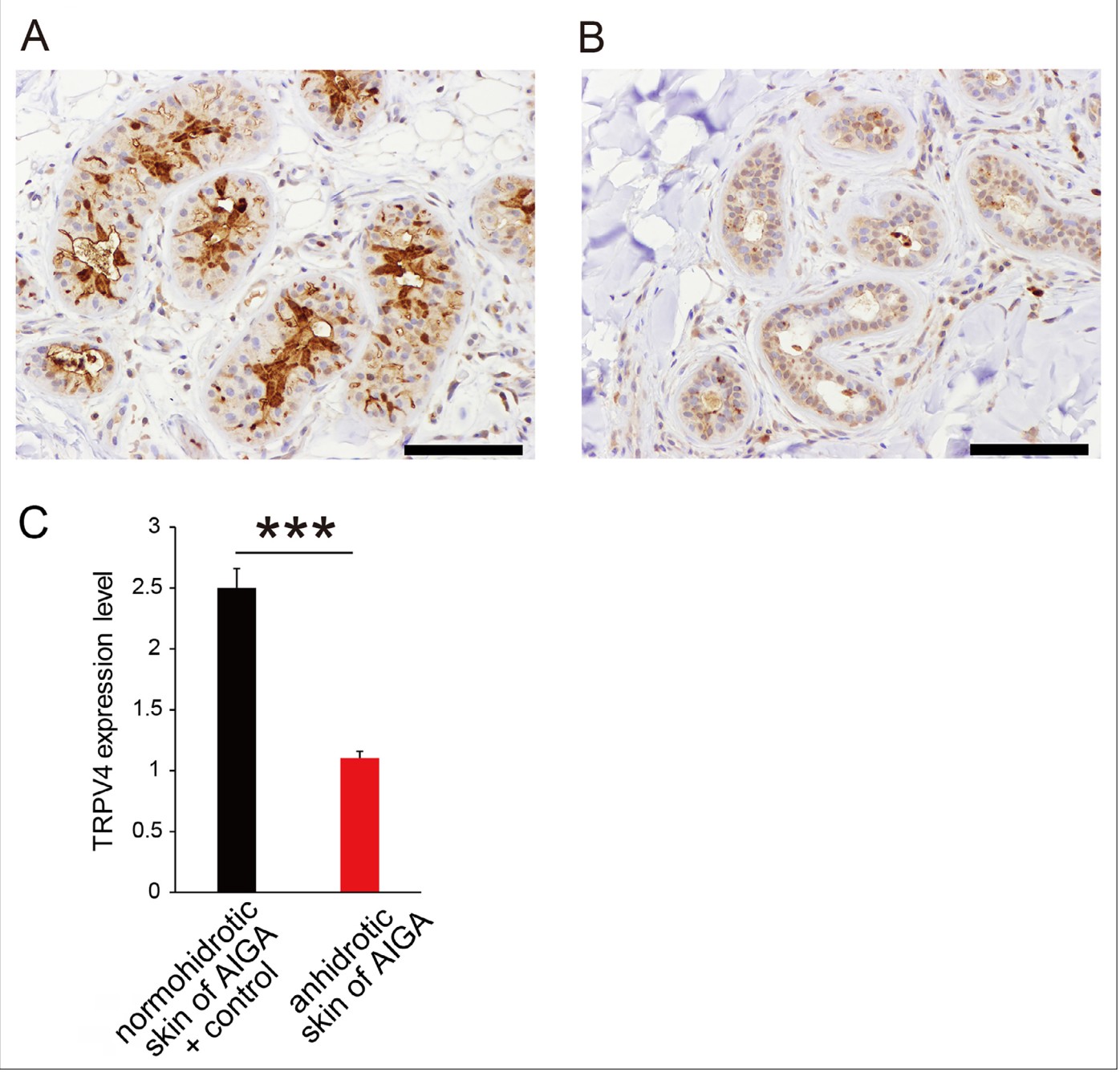

**Figure 5.** TRPV4 expression in human sweat glands. Representative TRPV4 staining in sweat glands from normohidrotic (**A**) and anhidrotic (**B**) skin from the same patient with acquired idiopathic generalized anhidrosis (AIGA). Scale bars: 50 μm. (**C**) Scored TRPV4 expression levels in normohidrotic skin from patients with AIGA and controls (melanocytic nevus) versus anhidrotic skin from patients with AIGA. *** p<0.001.

The online version of this article includes the following source data for figure 5:

**Source data 1.** Source data files for *Figure 5*.

Many TRP channels have high $Ca^{2+}$ permeability, suggesting that $Ca^{2+}$ entering cells through TRP channels in turn activates more $Ca^{2+}$-activated proteins, including other $Ca^{2+}$-activated ion channels such as $Ca^{2+}$-activated $K^+$ channels. This interaction could expand the importance of TRP channels in physiological functions, and complexes between TRP channels and $Ca^{2+}$-activated proteins would be novel targets for drug development.

# Materials and methods

**Key resources table**

| Reagent type (species) or resource | Designation | Source or reference | Identifiers | Additional information |
|---|---|---|---|---|
| Strain, strain background (*Escherichia coli*) | DH5a | Takara | 9057 | |
| Strain, strain background (mouse) | TRPV4KO | Dr. Makoto Suzuki (Jichi Medical University) | DOI: 10.1152/ajpcell.00559.2002 | |
| Strain, strain background (mouse) | TRPM8KO | Dr. Ardem Patapoutian (Scripps Research) | DOI: 10.1016/j. neuron.2007.02.024 | |
| Antibody | Anti-TRPV4 (guinea pig polyclonal) | In-house (Saga University) | DOI:10.1111/jre.12685 | IF (1:500) |
| Antibody | Anti-ANO1 (rabbit polyclonal) | Abcam | ab53213 | IF (1:100) |
| Antibody | Anti-AQP5 (rabbit polyclonal) | Millipore | 178615 | IF (1:200) |
| Antibody | Anti-CGRP (rabbit polyclonal) | Amersham International | RPN.1842 | IF (1:2000) |
| Antibody | Anti-cytokeratin 8 (CK8) (rat monoclonal) | Millipore | MABT329 | IF (1:100) |
| Antibody | Anti-TRPV4 antibody for human tissue | Abcam | Ab191580 | IF (1:100) |
| Recombinant DNA reagent | pcDNA3.1(+) (plasmid) | Invitrogen | V79020 | |
| Chemical compound, drug | GSK1016790A | Sigma-Aldrich | G0798 | |
| Chemical compound, drug | Menthol | Sigma-Aldrich | 63670 | |
| Chemical compound, drug | Ani9 | Sigma-Aldrich | SML1813 | |
| Software, algorithm | Origin Pro | Origin Pro | OriginLab | |

## Mice

Homozygous TRPV4-deficient (TRPV4KO) mice from Makoto Suzuki (Jichi Medical University) (*Mizuno et al., 2003*) and homozygous TRPM8-deficient (TRPM8KO) from Dr. David Julius were maintained under SPF conditions in a controlled environment (12 hr light/dark cycle with free access to food and water, 25°C, and 50–60% humidity). All procedures were approved by the Institutional Animal Care and Use Committee of the National Institute of Natural Sciences (approval no. 21A008) and carried out according to the National Institutes of Health and National Institute for Physiological Sciences guidelines. Because it was hard to prepare enough amount of TRPV4KO mice, we used a little wider distribution of mice age in the immunostaining (8- to 21-week-old mice) and did not use littermates. However, we back-crossed the KO mice to the commercially available WT mice more than 10 times, causing no difference in the genetic background between WT and TRPV4KO mice.

## Human ethics

Informed consent was obtained from all patients, and the study was approved by the Shinshu University Ethics Committee (approval no. 4073). Anhidrotic or hypohidrotic as well as normohidrotic skin samples taken from various sites were collected from 10 patients who were clinically diagnosed with AIGA using standard criteria set by the Japan AIGA study group (revised guideline for the diagnosis and treatment of AIGA in Japan) (*Munetsugu et al., 2017*).

## Chemicals

Collagenase A, trypsin from soybean, ionomycin calcium salt, Ach, carbachol, GSK1016790A (G0798), Ani9, and menthol were purchased from Sigma (St. Louis, MO).

## Isolation of sweat glands from mice

Dissected tips of digits and foot pads of mice were minced and incubated in 0.25 mg/mL liberase TL (Roche, 5401119001) for 45 min at 37°C with pipetting every 10 min. The digested tissue suspension was filtered through a 40 mm cell strainer, and the isolated sweat glands were retained in the filter. The collected sweat glands were seeded on Cell-Tak-coated glass slips and used for $Ca^{2+}$-imaging analysis after incubation at 37°C (>2 hr) in DMEM supplemented with 10% fetal bovine serum, penicillin-streptomycin, and GlutaMAX.

## Mouse immunostaining

Experiments were performed using 8- to 21-week-old male and female mice. Mice (n = 4 per group) were anesthetized with a combination of hydrochloric acid medetomidine (0.75 mg/kg; Kyoritsu Seiyaku, Tokyo, Japan), butorphanol (5 mg/kg; Meiji Seika Pharma, Tokyo, Japan), and midazolam (4 mg/kg, Maruishi, 21-3981), and perfused transcardially with heparinized PBS followed by 4% para-formaldehyde (PFA) in phosphate buffer (pH 7.4). The hind paw pad skin was dissected and post-fixed in 4% PFA for 3 hr at 4°C, cryoprotected with 20% sucrose overnight, and then embedded in OCT compound. For immunohistochemistry, 5-μm-thick frozen sections were made with a NX50 cryostat. Sections were permeabilized with 0.3% Triton-X100 in PBS for 10 min at room temperature and then incubated with a blocking solution, PBS supplemented with 0.3% Triton X-100, 1% bovine serum albumin, 0.05% sodium azide, and 5% normal donkey serum for 45 min at room temperature. Sections were then incubated overnight at 4°C with the primary antibodies: guinea pig anti-TRPV4 (2 μg/mL) (*Kitsuki et al., 2020*), rabbit anti-ANO1 (1:100, Abcam, ab53213), rabbit anti-AQP5 (1:200, Millipore, 178615), rabbit anti-CGRP (1:2000, Amersham International, RPN.1842), rat anti-cytokeratin 8 (CK8) (1:100, Millipore, MABT329). Then, sections were incubated for 1 hr at room temperature with the secondary antibodies: Alexa Fluor 488 donkey anti-guinea pig IgG, Alexa Fluor 555 donkey anti-rabbit IgG, and Alexa Fluor 647 donkey anti-rat IgG (all 1:200, Jackson ImmunoResearch Labs). F-actin was visualized with Phalloidin-iFluor 647 Reagent (1:1000). After immunostaining, sections were incubated for 5 min with 4',6-diamidino-2-phenylindole, dihydrochloride (DAPI, Dojindo, D523) and then mounted with PermaFluor (Thermo Fisher Scientific). Images were acquired using a BC43 or LSM800 instrument equipped with a Zeiss Axio Observer Z1 and an LSM 800 confocal unit with Airyscan module. For super-resolution imaging, images of optical 160-nm-thick slices were taken with a Plan Apochromat 63×/1.40 NA Oil DIC M27 objective. Images were processed with Airyscan processing in ZEN blue 3.5 software.

## Human immunostaining

Immunohistochemical analysis of formalin-fixed paraffin-embedded tissue sections (2–3-μm-thick) of anhidrotic and normohidrotic skin samples from 10 patients with AIGA was done. Except for the application of primary antibody (100×), all steps, including deparaffinization, blocking of internal peroxidase activity, unmasking of specific antigen, application of secondary antibody, detection of signals, and nuclear staining, were automatically performed using a Ventana auto-staining system. Skin samples with melanocytic nevus (n = 10) were used as a control.

## Iodine and starch test

Iodine starch assay was conducted using WT, TRPV4KO, and TRPM8KO male mice (8–14-week-old) at room temperature (25°C) or 35°C at 30–60% humidity. Mice were anesthetized by isoflurane inhalation and held in a prone position. Soles of the hind paw were painted with 3% iodine/EtOH and left to stand for 2 min to dry solvent. Thereafter, the same surfaces were painted with 10% starch/mineral oil, and positive signals of iodine-starch reaction (dark-blue color) were counted after 15 min for sweating without Ach stimulation. For the experiments with Ach injection, Ach (100 μM) in PBS (-) was subcutaneously administered after the starch painting and the signals were counted 2 min after Ach injection. For the analyses of Ani9 and menthol, Ani9 (10 μM in EtOH) or menthol (0.5% in EtOH) was topically applied to the soles 10 min before iodine-starch test.

## Calcium imaging

After loading with Fura-2 AM (5 μM, Invitrogen, F-1201), isolated sweat glands on coverslips were mounted in an open chamber and rinsed with standard bath solution containing (in mM) 140 NaCl, 5 KCl, 2 $MgCl_2$, 2 $CaCl_2$, 10 HEPES, and 10 glucose, pH 7.4. The intracellular free calcium concentration in isolated sweat glands was measured by dual-wavelength fura-2 microfluorometry with excitation at 340/380 nm and emission at 510 nm. The ratio image was calculated and acquired using the IP-Lab imaging processing system.

## Mouse climbing experiments

WT and TRPV4KO mice were allowed to acclimate for 1 day prior to recording in a cage containing a slippery slope made with vinyl. Mice were housed for 1 hr in the cage with the slope at 26–27°C and 35–50% humidity. Climbing and slipping behaviors were videotaped and analyzed.

## Quantifications and statistical analysis

Data are shown as mean ± sem. Statistical analysis was performed with Origin Pro 8. Student's *t*-test and two-way ANOVA with Dunnett's or Bonferroni's multiple-comparison tests were performed for comparisons. Values of p<0.05 indicate statistical significance.

## Acknowledgements

This work was supported by grants to MT from the Japan Society for the Promotion of Science (#20H05768, #21H02667, and #23H04943).

---

# Additional information

### Funding

| Funder | Grant reference number | Author |
|---|---|---|
| Japan Society for the Promotion of Science | 23H04943 | Makoto Tominaga |
| Japan Society for the Promotion of Science | 21H02667 | Makoto Tominaga |
| Japan Society for the Promotion of Science | 20H05768 | Makoto Tominaga |

The funders had no role in study design, data collection and interpretation, or the decision to submit the work for publication.

---

### Author contributions

Makiko Kashio, Conceptualization, Data curation, Formal analysis, Investigation, Methodology, Writing – original draft; Sandra Derouiche, Conceptualization, Data curation, Formal analysis, Investigation, Methodology; Reiko U Yoshimoto, Kenji Sano, Jing Lei, Conceptualization, Data curation, Formal analysis, Investigation; Mizuho A Kido, Data curation, Supervision, Investigation, Writing – review and editing; Makoto Tominaga, Conceptualization, Formal analysis, Funding acquisition, Validation, Methodology, Writing – original draft, Project administration, Writing – review and editing

### Author ORCIDs

Makiko Kashio http://orcid.org/0000-0001-6404-2339
Reiko U Yoshimoto https://orcid.org/0000-0001-8081-8926
Kenji Sano http://orcid.org/0000-0003-4312-2780
Mizuho A Kido https://orcid.org/0000-0002-9348-0706
Makoto Tominaga https://orcid.org/0000-0003-3111-3772

### Ethics

Human subjects: Informed consent was obtained from all patients, and the study was approved by the Shinshu University Ethics Committee (Approval No. 4073).

---

All procedures were approved by the Institutional Animal Care and Use Committee of the National Institute of Natural Sciences (Approval No. 21A008) and carried out according to the National Institutes of Health and National Institute for Physiological Sciences guidelines.

Joint Public Review: https://doi.org/10.7554/eLife.92993.3.sa1
Author response https://doi.org/10.7554/eLife.92993.3.sa2

---

# Additional files

## Supplementary files

• MDAR checklist

## Data availability

All data associated with this study are present in the paper or source data files.

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
