## [Editor Report · eLife assessment]

This **useful** studying implicates TRPV4 as a mediator of sweat, potentially based on TRPV4's expression and function on sweat glands. The data and methods are **solid**, with some limitations in terms of the approach. Overall, the work lends new insight into the physiological basis of sweating using data from mice and humans.

---

## [Referee Report · Joint Public Review]

In this study, Kashio et al examined the role of TRPV4 in regulating perspiration in mice. They find coexpression of TRPV4 with the chloride channel ANO1 and aquaporin 5, which implies possible coupling of heat sensing through TRPV4 to ion and water excretion through the latter channels. Calcium imaging of eccrine gland cells revealed that the TRPV4 agonist GSK101 activates these cells in WT mice, but not in TRPV4 KO. This effect is reduced with cold-stimulating menthol treatment. Temperature-dependent perspiration in mouse skin, either with passive heating or with ACh stimulation, was reduced in TRPV4 KO mice. Functional studies in mice - correlating the ability to climb a slippery slope to properly regulate skin moisture levels - reveal potential dysregulation of foot pad perspiration in TRPV4 KO mice, which had fewer successful climbing attempts. Lastly, a correlation of TRPV4 to hypohydrosis in humans was shown, as anhidrotic skin showed reduced levels of TRPV4 expression compared to normohidrotic or control skin.

Overall this is an interesting study on how TRPV4 regulates perspiration.

(1) The functional relationship between TRPV3 and ANO1 remains correlative.

(2) Littermate controls were not used, but TRPV4ko were backcrossed onto the WT strain.

(3) In general, the results support the authors' claims that TRPV4 activity is a necessary component of sweat gland secretion, which may have important implications for controlling perspiration; secretion from other glands where TRPV4 may be expressed remains a possibility given the lack of us of exocrine-specific knockouts.

---

## [Author Response]

The following is the authors’ response to the original reviews.

**Reviewer #1 (Public Review):**
(1) Measurement of secreted amylase could be seen as direct evidence of sweating, however, how to determine the causal relationship between climbing behavior and sweating? Friction force may also be reduced when there is too much fingertip moisture.

As the reviewer notes, measurement of secreted amylase can provide direct evidence of sweating, and we performed an iodine and starch reaction. Upon observing the involvement of TRPV4 in mouse foot pad perspiration, we then considered which type of behavioral analysis would be suitable to evaluate this perspiration. We agree with the reviewer’s point that friction force in the climbing test may be reduced by excessive sweating. However, we did not observe severe sweating in the absence of acetylcholine treatment. Accordingly, we interpreted that the increase in the climbing test failure rate for TRPV4KO mice could reflect the reduced friction force associated with the lack of TRPV4 activity.

(2) For the human skin immunostaining, did the author use the same TRPV4 antibody as used in the mouse staining? Did they validate the specificity of the antibody for the human TRPV4 channel?

We used different antibodies for human and mouse samples. Since commercially available anti-TRPV4 antibodies do not work well with mouse samples, we generated our own anti-TRPV4 antibody and validated its specificity.

(3) In lines 116-117, the authors tried to determine "the functional interaction of TRPV4 and ANO1 is involved in temperature-dependent sweating", however, they only used the TRPV4 ko mice and did not show any evidence supporting the relationship between TRPV4 and ANO1.

As the reviewer pointed out, based on the data presented in the original submission we cannot conclude that an interaction between TRPV4 and ANO1 is involved in perspiration. However, we think that the data for TRPV4KO mice presented in Figure 3 of the original version does indicate that TRPV4 is involved in perspiration. The finding that menthol and its related compounds, which inhibit the function of both TRPV4 and ANO1 (see our publication in *Scientific Reports* 7: 43132, 2017), blocked perspiration in both wild-type and TRPV4KO mice (original Figure 3C, D) indicates involvement of either TRPV4 or ANO1 in perspiration. In the revised version, we present results for additional iodine and starch reaction experiments using Ani9, a potent and specific ANO1 inhibitor. Ani9 drastically inhibited perspiration from mouse food pads both at 25 °C and 35 °C. Based on these collective results, we concluded that both TRPV4 and ANO1, likely acting as a complex, are involved in perspiration. We present the new data with Ani9 in the revised Figure 3E, F.

(4) Figure 3-4 is quite confusing. At 25°C, no sweating difference was observed between TRPV4 and wt mice (Fig 3A-3D), suggesting both Ach-induced sweating and basal sweating are TRPV4-independent at 25°C, however, the climbing test was done at 26-27 °C and the data showed a climbing deficit in TRPV4 ko mice. How to interpret the data is unclear.

Thank you for raising this point. In the iodine and starch reaction experiment, we observed no significant reduction in perspiration in the absence of acetylcholine at 25 °C, which is the same condition as in the climbing test, whereas we detected less perspiration for TRPV4KO mice. In a trial using additional mice, we detected significantly less perspiration under control conditions without acetylcholine at 25 °C, which is consistent with the results of the climbing test. We have added this new data to the revised Figure 3A, B.

(5) Were there any gender differences associated with sweating in mice? In Figure 3, the mouse number for behavior tests should be at least 5.

The TRPV4KO mice reproduced poorly and we were unable to obtain sufficient numbers of male and female mice to determine whether there were gender differences in sweating. However, according to the reviewer’s suggestion, and as mentioned above, we increased the number of experiments to obtain the results shown in the revised Figure 3. We did not a observe a significant difference in sweating with the larger sample size, which supports our conclusions.

(6) 8- to 21-week-old mice were used in the immunostaining, the time span is too long.

Given the difficulty in obtaining sufficient numbers of TRPV4KO mice, we used a somewhat wider age distribution to obtain samples for immunostaining. However, we did not observe age-dependent differences in immunostaining. We reference this point in the revised manuscript.

(7) The authors used homozygous TRPV4 ko mice for all experiments. What are control mice? Are they littermates of the TRPV4 ko mice?

We did not use littermates for our in vivo experiments because the TRPV4KO mice reproduced poorly and the litter sizes were small. However, we did backcross the KO mice to the commercially available wild-type mice more than ten times. As such, we expect that the wild-type and TRPV4KO mice will have similar genetic backgrounds. In addition, we have published multiple studies that have successfully used this method, which we think supports the reliability of our results for experiments involving mice.

**Reviewer #2 (Public Review):**
(1) The coexpression data needs additional controls. In the TRPV4 KO mice, there appears to be staining with the TRPV4 Ab in TRPV4 KO mice below the epidermis. This pattern appears similar to that of the location of the secretory coils of the sweat glands (Fig 1A). Is the co-staining the authors note later in Figure 1 also seen in TRPV4 KOs? This control should be shown, since the KO staining is not convincing that the Ab doesn't have off-target binding.

We thank the reviewer for raising these concerns about immunostaining. As the reviewer notes, in the low power image the signals appeared to be weak and punctate signals were present in the basal region of glandular cells. Although we did not identify immunohistochemical conditions that produced no signal, tissue sections from WT mice stained with anti-TRPV4 antibody showed conspicuous apical signals for the glandular cells facing lumen. Meanwhile, TRPV4KO tissues showed no signals at the apical region of the glandular cells, where the TRPV4-ANO1 interaction is expected to occur. We confirmed no trace signals in the TRPV4KO tissues in the immunoblotting.

(2) Are there any other markers besides CGRP for dark cells in mice to support the conclusion that mouse secretory cells have clear cell and dark cell properties?

We did not stain with other dark cell markers. Based on previous studies describing the differences between clear and dark cells in mouse eccrine glands, we think that dark and clear cells cannot be clearly discriminated, as we described in lines 93-96 of the Results. We identified secretory cells using CK8 and dark cells with CGRP, a marker of dark cells in human eccrine glands (Zancanaro et al. 1999 J Anat). Our result showed that CGRP immunostaining could not discriminate between clear and dark cells, which is consistent with a previous report showing that mouse secretory cells were assumed to be undifferentiated and primitive based on electron microscopic observation (Kurosumi et al. 1970 Arch Histol Jap).

(3) The authors utilize menthol (as a cooling stimulus) in several experiments. In the discussion, they interpret the effect of menthol as potentially disrupting TRPV4-ANO1 interactions independent of TRPM8. Yet, the role of TRPM8, such as in TRPM8 KO mice, is not evaluated in this study.

We performed the iodine and starch reaction experiments with TRPM8KO mice. In the TRPM8KO mice, the sweat spots did not differ from those seen for WT mice (p=0.63, t-test), and there was also a significant reduction in sweating with menthol treatment following acetylcholine stimulation that was similar to that seen for WT mice. These results would rule out the involvement of TRPM8 in a menthol-induced reduction in sweating. We have included this data in the revised Figure 3D.

(4) Along those lines, the authors suggest that menthol inhibits eccrine function, which might lead to a cooling sensation. But isn't the cooling sensation of sweating from evaporative cooling? In which case, inhibiting eccrine function may actually impair cooling sensations.

Menthol has a non-specific effect that activates TRPM8, TRPV3 and TRPA1, and inhibits TRPV1, TRPV4 and ANO1. Therefore, we did not carry out a climbing test with menthol in part because menthol-dependent TRPA1 activation decreased the propensity of the mice to climb. As the reviewer notes, TRPM8 activation following topical application of menthol may cause a cooling sensation elicited in sensory neurons beneath the skin. However, the comfortable cooling sensation could also be caused in part by decreased sweating. The relationship between a comfortable cooling sensation and less perspiration following menthol application may be difficult to determine, and we have mentioned this in the updated Discussion.

(5) The climbing assay is interesting and compelling. The authors note performing this under certain temperature and humidity conditions. Presumably, there is an optimal level of skin moisture, where skin that is too dry has less traction, but skin that is too wet may also have less traction. It would bolster this section of the study to perform this assay under hot conditions (perhaps TRPV4 KO mice, with impaired perspiration, would outperform WT mice with too much sweating?), or with pharmacologic intervention using TRPV4 agonists or antagonists to more rigorously evaluate whether this model correlates to TRPV4 function in the setting of different levels of perspiration.

We thank the reviewer for this suggestion. Upon detecting the involvement of TRPV4/ANO1 interaction in perspiration, we considered different behavioral analyses that can be performed to demonstrate whether the TRPV4/ANO1 interactions are involved in perspiration. As the reviewer suggested, there should be an optimal level of sweating. Therefore, we first set the room temperature at 26-27 °C and humidity at 35-50%. To our knowledge, this is the first demonstration of temperature-dependent sweating of mouse foot pads. In humans, palm sweating is often referred to as psychotic sweating that is known to be regulated by sympathetic nerve activity. Here we tested whether foot pad sweating might be related to friction force wherein sufficient amounts of sweating could increase the friction force and in turn increase the success rate for the climbing test using a vinyl-covered slippery slope that was selected based on several trials to determine the optimal surface material and slope angles. As the reviewer suggests, the success rates could be affected by multiple factors, and hot temperatures likely induce more sweating that could increase the success rates in the climbing test. We will need to carry out additional experiments that are beyond the scope of this study to examine these temperature-dependent effects. Generally, sweating is regulated by sympathetic nerve activity that occurs in response to increased brain neuron excitation. However, here we raise for the first time the possibility that sweating might be regulated by local temperature sensation mediated through TRPV4 that may be effective for fine-tuning of perspiration activity. We have updated the Discussion to reference this possibility.

(6) There are other studies (PMID 33085914, PMID 31216445) that have examined the role of TRPV4 in regulating perspiration. The presence of TRPV4 in eccrine glands is not a novel finding. Moreover, these studies noted that TRPV4 was not critical in regulating sweating in human subjects. These prior studies are in contradiction to the mouse data and the correlation to human anhidrotic skin in the present study. Neither of these studies is cited or discussed by the authors, but they should be.

We thank the reviewer for referencing these other studies concerning the possible involvement of TRPV4 in perspiration in humans. These studies focused on the vasodilating effects of TRPV4 and drew the conclusion that TRPV4 is not involved in sweating in humans, which is in contrast to our data for mice and humans. Multiple factors could explain the apparent difference between the two studies. For example, the parameters they examined differed from ours in that we assessed patients with AIGA, whereas the previous studies involved healthy volunteers. We have updated the Discussion to note the difference in the results of our and previous studies.

**Reviewer #3 (Public Review):**
(1) Figure 2: The calcium imaging-based approach shows average traces from 6 cells per genotype, but it was unclear if all acinar cells tested with this technique demonstrated TRPV4-mediated calcium influx, or if only a subset was presented.

“n = 6” does not indicate the number of cells, but rather 6 independent experiments that each had over 20 ROIs of sweat glands. We have clarified this point in the updated figure legend.

(2) Figure 4: The climbing behavioral test shows a significant reduction in climbing success rate in TRPV4-deficient mice. The authors ascribe this to a lack of hind paw 'traction' due to deficiencies in hind paw perspiration, but important controls and evidence that could rule out other potential confounds were not provided or cited.

As noted in our response to Comment 5 made by Reviewer #2, we spent considerable time identifying optimal conditions that would delineate success rates in the climbing experiments. We are confident that TRPV4KO mice had significantly lower success rates than WT mice, but there are various factors that could affect the experimental outcomes. We reference these factors in the updated Discussion.

(3) In general, the results support the authors' claims that TRPV4 activity is a necessary component of sweat gland secretion, which may have important implications for controlling perspiration as well as secretion from other glands where TRPV4 may be expressed.

As described above, the results we obtained in the climbing test can be affected by various factors. However, based on the consistency of the results obtained for the climbing test and the iodine and starch reaction assay, we think that our interpretation is correct. In terms of the involvement of TRPV4/ANO1 interactions in fluid secretion, we previously reported that the TRPV4/ANO1 complex is involved in cerebrospinal fluid secretion in the mouse choroid plexus (FASEB J. 2014) and in saliva and tear secretion in mouse salivary and lacrimal glands (FASEB J. 2018). Together, these findings suggest that this mechanism is common to water efflux from exocrine glands.

**Reviewer #1 (Recommendations For The Authors)**:(1) An exocrine gland-specific trpv4 knockout mouse should be used, as TRPV4 is also expressed by muscles, global knockout TRPV4 may affect the TRPV4-dependent muscle strength and reduce the climbing ability in mice.

As the reviewer suggests, use of mice with TRPV4 knockout specific to exocrine glands would be preferable to mice having global TRPV4 knockout given that TRPV4 is expressed in multiple tissues. We agree with this suggestion, but we do not currently have such mice in hand. However, as mentioned above, we have reported the involvement of theTRPV4/ANO1 interaction in cerebrospinal fluid secretion from the choroid plexus in mice (FASEB J. 28: 2238-2248, 2014), as well as saliva and tear secretion in mouse salivary and lacrimal glands (FASEB J. 32: 1841-1854, 2018.), suggesting that the TRPV4/ANO1 interaction could be widely involved in exocrine gland functions that involve water movement. We have updated the Discussion to reference this point.

(2) The authors showed Calcium imaging data that Menthol inhibits TRPV4-dependent calcium influx. However, it is well known that menthol induces the sensation of cooling by activating TRPM8. More evidence, including patch clamp recordings, should be done to verify the inhibition effects of menthol on TRPV4 and ANO1. Moreover, Fig 3E-3F could only suggest that menthol-induced cooling sensation may affect sweating but not the inhibition effect of menthol on TRPV4 and ANO1 channels.

We agree that more evidence including patch-clamp recordings can verify the inhibitory effects of menthol on TRPV4 and ANO1. We did not include such experiments here since we previously showed that menthol and related agents indeed inhibit TRPV4- and ANO1-mediated currents (Sci. Rep. 7: 43132, 2017). We now cite this paper in the revised version.

(3) Excepting the climbing test, are there any other better models to asses the sweating-related behaviors?

When we detected the involvement of TRPV4/ANO1 interactions in perspiration, we considered different types of behavioral analyses that could be used to demonstrate TRPV4/ANO1-dependent perspiration. We think that the climbing experiment is the best test, particularly since foot pads are one of the few regions on mice that is not covered by fur and thus amenable to evaluation of perspiration using an iodine and starch test.

**Reviewer #2 (Recommendations For The Authors):**
(1) I was confused by a section in the introduction on lines 59-60: How does Cl- efflux lead to the formation of a physical complex in cells with high intracellular Cl-? What is the physical complex? This seems like several disparate concepts combined together, which need to be clarified.

We apologize for the incomplete descriptions of several of our previous works. We have amended the Introduction section in the revised manuscript.

**Reviewer #3 (Recommendations For The Authors):**
(1) TRPV4 is expressed by multiple other cell types in the skin (keratinocytes, macrophages etc.) which may have an impact on peripheral sensory function. Is there evidence that TRPV4-deficient animals have relatively normal sensory acuity and/or proprioception? Such evidence would lend more credibility to the reported findings in the climbing test.

As the reviewer points out, TRPV4 is expressed by multiple other cell types in the skin. To date we have found that TRPV4KO mice show no differences in sensory functions compared to WT mice. Whether TRPV4 is involved in proprioception is unclear, based on both our own observation and those that appear in the literature, although TRPV4 is clearly activated by mechanical stimuli. We previously compared the mechanical sensitivity of TRPV4 and Piezo1 in bladder epithelial cells, and found that Piezo 1 shows much higher sensitivity relative to TRPV4 (J. Biol. Chem. 289: 16565-16575, 2014), which is consistent with the involvement of Piezo1, rather than TRPV4, in proprioception. Although TRPV4 is reported to be expressed in sensory neurons, we did not detect TRPV4-mediated responses in isolated rat and mouse DRG neurons, suggesting that TRPV4-positive sensory neurons are relatively rare.

(2) The methods section refers to loading entire sweat glands with Fura-2 dye for calcium imaging, but the figure legend refers to sweat gland acinar cells. Resolving this ambiguity would help readers to interpret the data.

We apologize for this error and have made an appropriate correction in the revised manuscript.

(3) Alternatively, could acute intraplantar injection of a TRPV4 antagonist (e.g. GSK205) in wild-type mice phenocopy the TRPV4-knockout mouse deficits, or could normal climbing behavior be restored in the TRPV4 knockout by adding artificial perspiration to their hindpaws?

We thank the reviewer for raising this interesting possibility and suggesting use of TRPV4 agonists or antagonists in the climbing tests. We agree that results of such an experiment would support the involvement of TRPV4 in sweating. We tried to do such experiments using injection of TRPV4 regulators into mouse hindpaws. However, the injections themselves appeared to impact climbing ability, perhaps in part due to painful sensations associated with the injection. Similarly, menthol injection appeared to reduce climbing activity, likely through pain sensations associated with TRPA1 activation. As such, we did not pursue these experiments.